# Prevalence of MCPyV, HPyV6, HPyV7 and TSPyV in Actinic Keratosis Biopsy Specimens

**DOI:** 10.3390/v14020427

**Published:** 2022-02-18

**Authors:** Carla Prezioso, Gabriele Brazzini, Sara Passerini, Carlotta Di Fabio, Terenzio Cosio, Sergio Bernardini, Elena Campione, Ugo Moens, Valeria Pietropaolo, Marco Ciotti

**Affiliations:** 1IRCSS San Raffaele Roma, Microbiology of Chronic Neuro-Degenerative Pathologies, 00163 Rome, Italy; carla.prezioso@uniroma1.it; 2Department of Public Health and Infectious Diseases, “Sapienza” University of Rome, 00185 Rome, Italy; gabriele.brazzini@uniroma1.it (G.B.); passerini.1915659@studenti.uniroma1.it (S.P.); valeria.pietropaolo@uniroma1.it (V.P.); 3Dermatology Unit, Department of Systems Medicine, Tor Vergata University Hospital, 00133 Rome, Italy; carlottadifabio@gmail.com (C.D.F.); terenziocosio@gmail.com (T.C.); campioneelena@hotmail.com (E.C.); 4Department of Experimental Medicine, Tor Vergata University of Rome, 00133 Rome, Italy; bernardini@med.uniroma2.it; 5Department of Medical Biology, Faculty of Health Sciences, University of Tromsø—The Arctic University of Norway, 9037 Tromsø, Norway; ugo.moens@uit.no; 6Virology Unit, Polyclinic Tor Vergata Foundation, Viale Oxford 81, 00133 Rome, Italy

**Keywords:** human polyomaviruses, skin, actinic keratosis, skin cancers, PCR, sequencing

## Abstract

To date, 14 human polyomaviruses (HPyVs) have been identified using high-throughput technologies. Among them, MCPyV, HPyV6, HPyV7 and TSPyV present a skin tropism, but a causal role in skin diseases has been established only for MCPyV as a causative agent of Merkel cell carcinoma (MCC) and TSPyV as an etiological agent of Trichodysplasia Spinulosa (TS). In the search for a possible role for cutaneous HPyVs in the development of skin malignant lesions, we investigated the prevalence of MCPyV, HPyV6, HPyV7 and TSPyV in actinic keratosis (AK), a premalignant skin lesion that has the potential to progress towards a squamous cell carcinoma (SCC). One skin lesion and one non-lesion skin from nine affected individuals were analyzed by qualitative PCR. MCPyV was detected in 9 out of 9 lesion biopsies and 6 out of 8 non-lesion biopsies. HPyV6 was detected only in healthy skin, while HPyV7 and TSPyV were not detected in any skin sample. These findings argue against a possible role of cutaneous HPyVs in AK. However, considering the small sample size analyzed, a definitive conclusion cannot be drawn. Longitudinal studies on large cohorts are warranted.

## 1. Introduction

Human Polyomaviruses (HPyVs) are ubiquitous viruses with high seroprevalences in the general population [1,2]. They contain a circular double-stranded DNA genome of approximately 5.0 kbp [1,2] encoding regulatory proteins and structural proteins. The major regulatory proteins are the large tumor antigen (LT) and the small tumor antigen (sT), while at least two structural proteins (VP1 and VP2) form the capsid. The regulatory proteins are expressed early during infection and participate in viral replication and viral transcription, while the structural proteins are expressed later in the infection cycle [1]. HPyVs may encode additional regulatory and structural proteins (e.g., ALTO, VP3, VP4, agnoprotein) [1].

After asymptomatic primary infection, HPyVs can establish a latent infection in different organs and tissues such as the brain, urogenital tract, and skin [1,2]. BKPyV and JCPyV represent the first two PyVs isolated from humans. They were both uncovered in 1971, and, under different conditions of immunosuppression, JCPyV can cause Progressive Multifocal Leukoencephalopathy (PML), whereas BKPyV can progress to BKPyV-associated nephropathy (BKVN) and eventually graft loss in kidney transplant patients [1,3].

Thanks to the use of high-throughput technologies, in recent years, a rapid expansion of the *Polyomaviridae* family was observed. To date, 14 HPyVs have been identified and characterized in terms of genome structure and related disease [3].

Several HPyVs have been detected in the skin, including MCPyV, HPyV6, HPyV7, TSPyV, HPyV9, HPyV10, STLPyV, and NJPyV-2013 [4]. MCPyV, HPyV6, HPyV7, and HPyV10 seem to be the most common HPyVs in healthy skin, with prevalences of approximately 67%, 17–30%, 10–15%, and 15%, respectively. STLPyV, TSPyV, and HPyV9 are less common with prevalences of 8%, 1%, and 1%, respectively [5,6,7]. NJPyV-2013 has only been detected in necrotic plaques on the hand, face, and scalp of a pancreatic transplant patient [8]. The association with human disease has been established only for Merkel cell polyomavirus (MCPyV), considered the etiological agent of Merkel cell carcinoma (MCC) [9,10], for HPyV6 and HPyV7, associated with pruritic rash and dyskeratotic dermatoses [11,12,13,14,15], and for Trichodysplasia Spinulosa-associated PyV (TSPyV), identified as the causative agent of Trichodysplasia Spinulosa (TS) [16].

In recent years, several viruses have been proposed as playing a role in the development of skin cancers [17,18], although among HPyVs, a viral-causal role has been established only for MCC [10]. It is therefore reasonable to determine whether a virus or viruses, such as cutaneous HPyVs, could contribute to the risk of the development of skin cancers. Given this background, our study aimed at investigating the prevalence of MCPyV, HPyV6, HPyV7, and TSPyV in skin biopsies from normal-appearing skin and from the lesion skin of patients affected by actinic keratosis (AK), a well-established pre-cancerous skin lesion that has the potential to progress to squamous cell carcinoma (SCC), especially in immune-suppressed patients [19].

AK is a chronic skin disease characterized by multiple clinical and subclinical lesions present in wide areas of sun-exposed skin known as field cancerization. Lesions require treatment because of their potential to transform into SCC. A high degree of UVB exposure, increasing age, male gender, and fair skin are risk factors [20]. Actually, UVB causes direct damage to DNA, producing pyrimidine dimers and suppressing the protective role of p53. The stepwise progression of AK, with an increased expression of anti-apoptotic Bcl-2, leads to SCC [20]. Because AK is typical for immunosuppressed subjects, like the skin pathologies described above (MCC, pruritic rash, dyskeratotic dermatoses and TS) where HPyVs play a role, the investigation of the possible role of MCPyV, HPyV6, HPyV7, and TSPyV in the development of AK could be significant [21].

## 2. Materials and Methods

### 2.1. Patients and Samples

Fresh tissue samples from actinic keratosis (AK) biopsies were collected from nine male patients admitted to the Dermatology Clinic of Tor Vergata University Hospital, Rome, Italy, for virological analysis.

Lesion and non-lesion skin were obtained from eight patients, whereas for one patient the only lesional part was collected. Biopsies were collected from March 2021 to October 2021 and analyzed in November 2021.

The study was approved by the local Ethic Committee of the University Hospital Tor Vergata (Rome, Italy) (protocol number 0015440/2019, 1 July 2019) and patients’ informed consent was obtained. Demographic and clinical characteristics are presented in Table 1.

### 2.2. DNA Extraction

DNA extraction was performed using Quick-DNA FFPE Miniprep (Zymo Research, Irvine, CA) according to the manufacturer’s instructions. The extracted nucleic acids were eluted in a final volume of 50 μL, and DNA was evaluated for its PCR suitability by amplifying the β-globin gene sequences [22].

### 2.3. Qualitative HPyVs PCR

Positive HPyVs DNA samples were subjected to qualitative PCR with different specific primer pairs mapping VP1 and LT regions of the genome and were subsequently sequenced, following published protocol.

Specifically, for the amplification of MCPyV DNA, primers targeting the highly variable BC loop of VP1 gene were used. The first round of amplification was performed using the primer pair 5′-TGCAAATCCAGAGGTTCTCC-3′ and 5′-AAAACACCCAAAAGGCAA TG-3′, generating a 494-bp segment. The second PCR was performed using the primers 5′-ATATTGCCTCCCACATCTGC-3′ and 5′-TGCCCTAAT GTTGCCTCAGT-3′, producing a fragment of 307 bp. The plasmid pMCV-R17a containing the complete genome of MCPyV (catalog No. 24729; Addgene, Cambridge, MA, USA) was used as a positive control, and distilled water was used as a negative control. The PCR conditions included denaturation for 5 min at 94 °C, followed by 35 cycles of 30 s at 94 °C, 30 s at 60 °C, and 30 s at 72 °C, and a final extension step of 7 min at 72 °C [23,24].

For HPyV6, the following primers were used to amplify a 299 bp fragment of LT: HPyV6 F: CAATGCATCACTACCTGGAC (nucleotides 4316–4335 in isolate 601a; HM011558) and HPyV6 R: GTTTGGGATTTCCGTTTGTG (nucleotides 4595–4614 in isolate 601a; HM011558). For HPyV7, a 344 bp fragment of LT was amplified using the primers HPyV7, F: CACGCAGGGCTTCCATATGG (nucleotides 4267–4282 in isolate 713a; HM011586) and HPyV7 R: GGTTTAAGAGCCTGCTGTTG-3′ (nucleotides 4591–4610 in isolate 713a; HM011586) [25]. The PCR conditions were 40 cycles of 30 s at 90 °C, 30 s at 55 °C, and 1 min at 72 °C. PCR was performed with Accustart II PCR SuperMix (Quantabio, Beverly, MA, USA). The plasmids pHPyV6-607a and pHPyV7-713a (Addgene, Watertown, MA, USA) were used as positive controls and to test the sensitivity of our PCR.

For TSPyV DNA amplification, VP1 internal primers (forward, GTGGGGAACCCCTAGAACTC; reverse TCTCCATCTTTCCCAACCAG) were used with the following thermal profile: denaturation at 95 °C for 30 s, primer annealing at 55 °C for 30 s, elongation at 72 °C for 30 s, obtaining a specific fragment of 370 bp [26]. The plasmid pUC19-TSV (Addgene, Watertown, MA, USA) was used as a positive control [16].

Ten microliters of amplification products were analyzed by electrophoresis in 2% agarose gels stained with ethidium bromide and observed under UV light.

### 2.4. Viral DNA Sequencing and Sequence Alignment

After purification of the PCR products by the MinElute PCR Purification Kit (QIAGEN, Milan, Italy), sequencing was performed in a dedicated facility (Bio-Fab Research, Roma, Italy). Sequences were generated using the Big Dye Terminator Sequencing method (Life Technologies, Carlstadt, CA, USA) on the ABI 3730 sequencer (Life Technologies), and analyzed with the Sequencing Analysis 5.2 software (Life Technologies). The obtained sequences were compared to reference sequences deposited in GenBank. Sequence alignments were performed with ClustalW2 at the European Molecular Biology Laboratory-European Bioinformatics Institute (EMBL-EBI) website using default parameters [27].

### 2.5. Statistical Analysis

MCPyV, HPyV6, HPyV7, and TSPyV detection was summarized by counts and proportions. If continuous variables were normally distributed, they were expressed as mean ± standard deviation; if not, they were expressed by median and range. The χ^2^ test was performed to evaluate differences in the viral detection, and the Mann–Whitney U-test for non-normally distributed continuous variables was applied to analyze differences between patients. A *p* value < 0.05 was considered statistically significant.

## 3. Results and Discussion

The presence of MCPyV, HPyV6, HPyV7, and TSPyV DNA in lesion and non-lesion skin sections was determined by qualitative PCR.

MCPyV DNA was analyzed using primers amplifying the VP1 region of the genome (307 bp). As shown in Table 2, MCPyV DNA was found in all lesion tissue samples (9/9; 100%) and in 6/8 (75%) non-lesion biopsy samples. This finding confirms that MCPyV is part of cutaneous microbiota and is commonly isolated from skin samples of adult subjects.

A qualitative PCR analysis using primers amplifying the LT region of the genome was performed to find HPyV6 and HPyV7. HPyV6 was found only in 1/8 (75%) of healthy tissue samples, while HPyV7 was not found in any samples (Table 2). These results seem to confirm that HPyV6 is, like MCPyV, part of the skin microbiota, whereas the absence of HPyV6 and HPyV7 in AK samples could confirm that these two HPyVs are not involved in cutaneous malignancies, as reported in previous studies [6,19].

Finally, the presence of TSPyV DNA was tested using primers targeting the VP1 sequence. According to Chen and colleagues [28], this virus associated with TS reaches a seroprevalence of 70% in adults. Nevertheless, in contrast to its high prevalence, in this study none of the tested biopsies were found positive for TSPyV.

Positive samples were sequenced in the dedicated facility BioFab. The obtained sequences were aligned and compared to the reference sequence of the relative HPyV. For MCPyV, the reference sequence from isolate R17b [GenBank: HM011556.1] was used, while for HPyV6 the reference sequence from isolate F-SKP61079 was used [GenBank: LC309185.1].

All biopsy samples that were found positive for MCPyV showed a wild-type VP1 sequence (Figure 1). P1, found positive for HPyV6, showed a LT sequence with a two-point mutation when compared to the reference one: the first was 73A > G, and the second was 135A > G (Figure 2).

## 4. Conclusions

This preliminary study speaks against a possible causal role of the four human polyomaviruses investigated in AK. MCPyV was the only polyomavirus found in AK lesions, but it was detected at a slightly higher frequency than non-lesion skin. In a study carried out on a cohort of liver transplant patients with pre-malignant and malignant skin lesions [19], MCPyV was found in a patient with AK, while Scola et al. [29] detected MCPyV in 19% (6/31) of investigated AK lesions.

In our study, HPyV6 was only detected in non-lesion skin, unlike Scola et al. [29], who found HPyV6 in 3% (1/31) of AK lesions. Finally, HPyV7 and TSV were detected in neither lesion nor non-lesion biopsies, in agreement with Scola’s work.

The strength of this work is that the presence of DNA from the four most common dermatropic HPyVs was examined in lesion and non-lesion skin biopsies by the highly sensitive PCR technique and that the amplified products were sequenced. A caveat of this study is the limited number of individuals that were investigated. Moreover, there was a bias in the gender because all patients were males.

Future research to establish a possible role of MCPyV, HPyV6, HPyV7, and TSPyV in AK should include an investigation of the genome copy number, the state of the viral genome, and the expression of viral genes. In addition, longitudinal studies are needed in order to establish whether persistent infection is required and whether sequence variations in the viral genomes occur. Considering the small sample size that was analyzed in the present study and in the studies cited above, no definitive conclusions can be drawn.

## Figures and Tables

**Figure 1 viruses-14-00427-f001:**
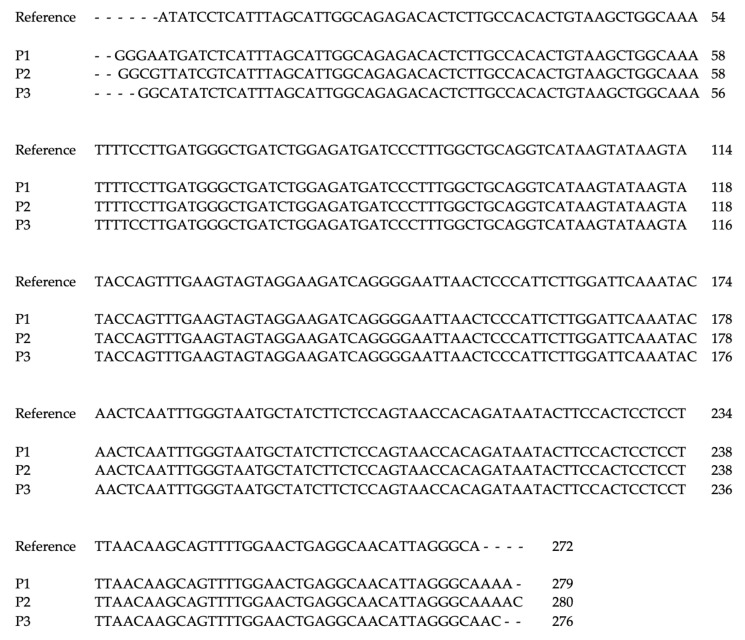
Multiple alignment of MCPyV-VP1 sequence found in biopsy samples, compared to reference sequence from isolate R17b [GenBank: HM011556.1]. P1: Patient 1, P2: Patient 2, P3: Patient 3.

**Figure 2 viruses-14-00427-f002:**
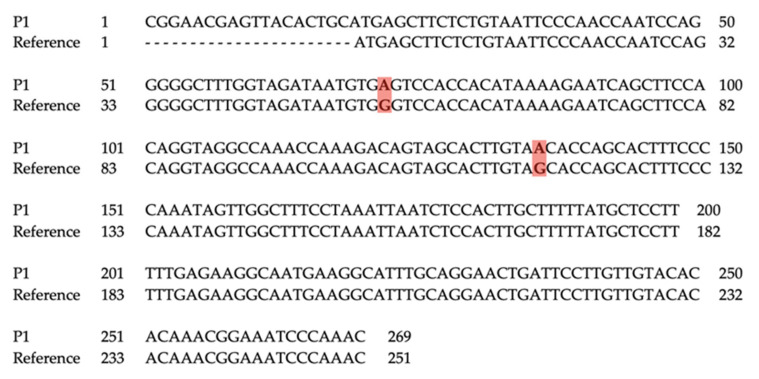
Pairwise alignment of HPyV6-LT sequence from healthy tissue sample of patient P1, compared to reference sequence from isolate F-SKP61079 [GenBank: LC309185.1]. P1: Patient 1.

**Table 1 viruses-14-00427-t001:** Demographic and clinical characteristics of AK patients.

Patients	Sex	Age	Diagnosis	Lesion Biopsy	Non-Lesion Biopsy
1	M	70	AK	+	+
2	M	83	AK	+	+
3	M	80	AK	+	−
4	M	81	AK	+	+
5	M	82	AK	+	+
6	M	75	AK	+	+
7	M	86	AK	+	+
8	M	78	AK	+	+
9	M	72	AK	+	+

Patients: 1–9; M: Male, AK: Actinic Keratosis, +: lesion and non-lesion biopsy from the patient, −: absence of non-lesion biopsy from the patient.

**Table 2 viruses-14-00427-t002:** PCR results.

Samples	n	MCPyV	HPyV6	HPyV7	TSPyV
Lesion biopsy	9	9 (100%)	0 (0%)	0 (0%)	0 (0%)
Non-lesion biopsy	8	5 (62.5%)	1 (12.5%)	0 (0%)	0 (0%)

MCPyV: Merkel-Cell Polyomavirus; HPyV: Human Polyomavirus; TSPyV: Trichodysplasia Spinulosa-associated Polyomavirus.

## Data Availability

Data is contained within the article.

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
