# Peer review of "Prevalence of MCPyV, HPyV6, HPyV7 and TSPyV in Actinic Keratosis Biopsy Specimens"

_viruses, 2022, doi:10.3390/v14020427_

Round 1

Reviewer 1 Report

The paper, submitted in the form of a communication, from Prezioso and colleagues focuses on the prevalence of four human polyomaviruses in skin biopsies taken from nine patients with actinic keratoses (AK).  These skin presentations typically occur in older patients and most of the patients with one exception were 70 years or older.  One patient was 25. All were male. It’s unclear to this reviewer why females were excluded in the study. Men are more likely to be affected but females certainly do present with AK.

The data are clear, controls are in place, and one can conclude that MCPyV is certainly prevalent in lesions and non-lesions but the other four are rarely detected in skin. The caveat is that the sample size is very low, n=9. The authors are aware of this and suggest that larger studies are needed. This reviewer agrees but also suggests a series of smaller studies accumulated over time by multiple groups could prove useful.

In summary this is a short focused paper that provides important observations although they are limited in scope given the small sample size and the lack of inclusion of female subjects.

Author Response

The paper, submitted in the form of a communication, from Prezioso and colleagues focuses on the prevalence of four human polyomaviruses in skin biopsies taken from nine patients with actinic keratoses (AK).  These skin presentations typically occur in older patients and most of the patients with one exception were 70 years or older.  One patient was 25. All were male. It’s unclear to this reviewer why females were excluded in the study. Men are more likely to be affected but females certainly do present with AK.

The data are clear, controls are in place, and one can conclude that MCPyV is certainly prevalent in lesions and non-lesions but the other four are rarely detected in skin. The caveat is that the sample size is very low, n=9. The authors are aware of this and suggest that larger studies are needed. This reviewer agrees but also suggests a series of smaller studies accumulated over time by multiple groups could prove useful.

In summary this is a short focused paper that provides important observations although they are limited in scope given the small sample size and the lack of inclusion of female subjects.

Response: We completely agree with the reviewer. Therefore, we have added following sentence in the abstract:

Longitudinal studies on large cohorts are warranted.

We also edited part of the Conclusions indicating the strength and limitations of this study (see reviewer 2) and addressing future research necessary to understand the role, if any, of MCPyV, HPyV6, HPyV7 and TSPyV in AK.

We did not exclude females from our study. It was only by chance that all 9 patients enrolled were males. All patients were 70 years or older. The age 25 was a typo that has been corrected.

Reviewer 2 Report

Authors aim to explore whether the prevalence of skin polyomaviruses, including MCPyV, HPyV6, HPyV7 and TSPyV, is different in skin lesions of actinic keratosis than non-skin lesions of the same participants. Although their research hypothesis is interesting, the study design is very poor and they were able to include only 9 patients.

Comment 1. Abstract: authors should include the size of the study sample (how many cases, how many controls).

Comment2 . There is no need for SCC abbreviation in the introduction.

Comment 3. Abstract. The correct abbreviation for human polyomaviruses is HPyVs not HpyVs

Comment 4. Introduction. It is better if you keep the introduction focused on the skin polyomaviruses, their skin tropism and their ability to cause or not disease and under which circumstances. Here is a really nice review by Sheu et al. “Polyomaviruses of the skin: integrating molecular and clinical advances in an emerging class of viruses” and by Hashida et al. Prevalence and Viral Loads of Cutaneous Human Polyomaviruses in the Skin of Patients With Chronic Inflammatory Skin Diseases 

Comment 5 It would also be interesting to describe a little bit more on actinic keatosis, e.g. aetiology , risk factors, progression etc. Are there any previous studies examining this association?

Comment 6. “Anyway, the clinical relevance for the majority of them remains an open question. “ please delete this sentence from introduction.

Comment 7. HPyV6 and HPyV7 are both linked to pruritic skin eruptions. Please include this association in the introduction.

Comment 8. Methods. Please provide more information on your study sample. Why only nine? Why only male? Please include dates that these biopsies were collected and when analysed. How did you define actinic ceratosis? Were skin biopsies from lesion and non-lesion areas obtained the same date? If yes is there any risk of contamination? Also comment why did you choose for comparison non-lesion skin biopsies from the same participant rather that biopsies form a healthy participant? Authors should discuss all these limitations in the discussion section as well. Also report why you did not test for the other skin polyomaviruses.

Comment 9. Statistical analysis. Provide the abbreviation of SD. What do you mean by _2 test? Did you compare patients or lesions? Your two groups of comparison are not independent because you have two biopsies (one from normal skin and one with actinic ceratosis) by each participant so the statistics you will use dependent on that. Moreover, I don’t see any comparisons or reporting of p-values in the result section.

Comment 10. Results. “This finding confirms that MCPyV is part of cutaneous microbiota and is commonly isolated from skin samples of adult subjects.”  And “These results seem to confirm that HPyV6 is, like MCPyV, part of the skin microbiota whereas, the absence of HPyV6 and HPyV7 in AK samples could confirm that these two HPyVs are not involved in cutaneous malignancies, as reported in previous studies [37,38].” And “According to Chen and colleagues [39], this virus associated to TS reaches a seroprevalence of 70% in adults.” All these are interpretations/comments of your results. Better keep this for discussion.

Comment 11. Results. Table 2. Present results also by participant. There should be people discordant to MCPyV and HPyV6 viruses. Was the severity of actinic keratosis different? Were they older people?

Comment 12. Discussion. Authors provide a very brief discussion of their research findings. Strengths and limitations are not discussed at all. There are also alternative explanations related to methodology of their findings.

Author Response

Authors aim to explore whether the prevalence of skin polyomaviruses, including MCPyV, HPyV6, HPyV7 and TSPyV, is different in skin lesions of actinic keratosis than non-skin lesions of the same participants. Although their research hypothesis is interesting, the study design is very poor and they were able to include only 9 patients.

Comment 1. Abstract: authors should include the size of the study sample (how many cases, how many controls).

Response: we have changed the abstract as follows:

One skin lesion and one non-lesion of nine affected individuals were analyzed by qualitative PCR.

Comment 2. There is no need for SCC abbreviation in the introduction.

Response: we agree with the reviewer that SCC in a universal abbreviation in the field of cancer, but the instruction of the journal requires that: “abbreviations should be defined the first time they appear in each of three sections: the abstract; the main text; the first figure or table. When defined for the first time, the acronym/abbreviation/initialism should be added in parentheses after the written-out form.”

Hence, we have kept the abbreviation in the introduction.

Comment 3. Abstract. The correct abbreviation for human polyomaviruses is HPyVs not HpyVs

Response: we have corrected this typo.

Comment 4. Introduction. It is better if you keep the introduction focused on the skin polyomaviruses, their skin tropism and their ability to cause or not disease and under which circumstances. Here is a really nice review by Sheu et al. “Polyomaviruses of the skin: integrating molecular and clinical advances in an emerging class of viruses” and by Hashida et al. Prevalence and Viral Loads of Cutaneous Human Polyomaviruses in the Skin of Patients With Chronic Inflammatory Skin Diseases 

Response: We have changed the introduction focusing on the skin polyomaviruses and their possible involvement in malignant and non-malignant skin diseases.

Comment 5 It would also be interesting to describe a little bit more on actinic keratosis, e.g. aetiology, risk factors, progression etc. Are there any previous studies examining this association?

Response:  We added more information on actinic keratosis including aetiology, risk factors and progression, and cited some works where the association between human polyomaviruses and AK was investigated: Scola et al Brit. J. Dermatol. 2012, 167, 1315-1320 ([30]) and Wang et al. Transpl. Int. 2019, 32, 516–522 ([28[).

Comment 6. “Anyway, the clinical relevance for the majority of them remains an open question. “please delete this sentence from introduction.

Response: we have deleted this sentence.

Comment 7. HPyV6 and HPyV7 are both linked to pruritic skin eruptions. Please include this association in the introduction.

Response: We thank the reviewer for this important comment. We included the association in the introduction as suggested. So, now you find the following text:

HPyV6 and HPyV7 have been linked to pruritic skin eruptions [11-15].

  1. Ho, J.; Jedrych, J.J.; Feng, H.; Natalie, A.A.; Grandinetti, L.; Mirvish, E.; Crespo, M.M.; Yadav, D.; Fasanella, K.E.; Proksell, S.; et al. Human polyomavirus 7-associated pruritic rash and viremia in transplant recipients. J. Infect. Dis. 2015, 211, 1560–1565.
  2. Nguyen, K.D.; Lee, E.E.; Yue, Y.; Stork, J.; Pock, L.; North, J.P.; Vandergriff, T.; Cockerell, C.; Hosler, G.A.; Pastrana, D.V.; et al. Human polyomavirus 6 and 7 are associated with pruritic and dyskeratotic dermatoses. J. Am. Acad. Dermatol. 2017, 76, 932–940.e3.
  3. Smith, S.D.B.; Erdag, G.; Cuda, J.D.; Rangwala, S.; Girardi, N.; Bibee, K.; Orens, J.B., Prono, M.D.; Toptan, T.; Loss, M.J. Treatment of human polyomavirus-7-associated rash and pruritus with topical cidofovir in a lung transplant patient: Case report and literature review. Transpl. Infect. Dis. 2018, 20, doi: 10.1111/tid.12793.
  4. Rosenstein, R.K.; Pastrana, D.V.; Starrett, G.J.; Sapio, M.R.; Hill, N.T.; Jo, J.H.; Lee, C.R.; Iadarola, M.J.; Buck, C.B.; Kong, H.H.; et al. Host-Pathogen Interactions in Human Polyomavirus 7‒Associated Pruritic Skin Eruption. J. Invest. Dermatol. 2021, 141, 1344-1348.e8.
  5. Kwan, K.; Sears, S.; Callen, J.; Rady, P.; Tyring, S.; Bahrami, S.; Huelsman, M.; Malone, J. Keratotic spines in a patient with pruritic and dyskeratotic dermatosis: A new clinical finding. JAAD Case Rep. 2020, 7, 103–106. 

Comment 8. Methods. Please provide more information on your study sample. Why only nine? Why only male? Please include dates that these biopsies were collected and when analysed. How did you define actinic keratosis? Were skin biopsies from lesion and non-lesion areas obtained the same date? If yes is there any risk of contamination? Also comment why did you choose for comparison non-lesion skin biopsies from the same participant rather that biopsies form a healthy participant? Authors should discuss all these limitations in the discussion section as well. Also report why you did not test for the other skin polyomaviruses.

Response: This is an ongoing study, and at the time of this Communication we enrolled only nine patients. It was only by chance that all patients enrolled were males. Our study design does not exclude female patients that most likely will be enrolled in future.

Biopsies were collected from March 2021 to October 2021 and analyzed in November 2021.This information is now reported in material and methods.

Actinic keratosis was defined in accordance with the macroscopic and dermatoscopic characteristics at the dedicated dermatological clinic. Diagnosis was confirmed by histological examination.

Biopsies from lesion and non-lesion areas were obtained the same day following the following procedure. The first dermatologist performed the biopsy in the area of clinically injured skin and placed the removed tissue in a sterile container. A second dermatologist using his own surgical instruments performed the biopsy in a non-lesion site in order to obtain healthy skin as control. Also, this second biopsy was placed in a sterile container and sent to the virology lab for PCR investigations.

We decide to collect for comparison non-lesion biopsies from the same participant rather than from a healthy control because biopsy is an invasive procedure and people are reluctant to undergo biopsy if not necessary. Furthermore, the Ethical Committee wouldn’t have granted approval. We also believe that a non-lesion biopsy from the same patient is a better control than a non-lesion biopsy from a healthy subject if you want to study the possible role of HPyVs in AK.

About the limitations of the study, see comment 12.

We did not test other skin polyomaviruses such as HPyV9, HPyV10, STLPyV and NJPyV (=HPyV13) because they are not frequently detected in the skin (for recent review see Sheu et al. Br. J. Dermatol. 2019; 180:1302-1311 and Bopp et al. Front. Microbiol. 2021; 12:740947). Instead, MCPyV, HPyV6 and HPyV7 are most common in the skin and have been associated with disease, whereas TSPyV is less common in healthy skin, but has been associated with disease; therefore, we focused on these 4 HPyVs. HPyV9, HPyV10 and STLPyV have not been convincingly associated with any disease, including skin disease.  Moreover, the prevalence of HPyV9 in normal skin was found to be low (<1% of the tested samples in the study by Bopp et al.). To the best of our knowledge, NJPyV has only been detected in necrotic plaques on the hand, face and scalp of a pancreatic transplant patient. Therefore, HPyV9, HPyV10, STLPyV, and NJPyV were not considered in our study.

Comment 9. Statistical analysis. Provide the abbreviation of SD. What do you mean by _2 test? Did you compare patients or lesions? Your two groups of comparison are not independent because you have two biopsies (one from normal skin and one with actinic keratosis) by each participant so the statistics you will use dependent on that. Moreover, I don’t see any comparisons or reporting of p-values in the result section.

Response: We have replaced SD by standard deviation. Formatting the paper resulted in the loss of c in the c2 test. We have corrected this mistake.

Comment 10. Results. “This finding confirms that MCPyV is part of cutaneous microbiota and is commonly isolated from skin samples of adult subjects.”  And “These results seem to confirm that HPyV6 is, like MCPyV, part of the skin microbiota whereas, the absence of HPyV6 and HPyV7 in AK samples could confirm that these two HPyVs are not involved in cutaneous malignancies, as reported in previous studies [37,38].” And “According to Chen and colleagues [39], this virus associated to TS reaches a seroprevalence of 70% in adults.” All these are interpretations/comments of your results. Better keep this for discussion.

Response: We agree with the reviewer, but since our short communication has a combined Results and Discussion section, interpretation of the results can be immediately discussed and not addressed in a separate Discussion section.

Comment 11. Results. Table 2. Present results also by participant. There should be people discordant to MCPyV and HPyV6 viruses. Was the severity of actinic keratosis different? Were they older people?

The severity of actinic keratosis was similar among the subjects enrolled: clinical grading I in 7/9, and clinical grading II in 2/9. All patients presented multiple actinic keratosis lesions and the age of the only patient with positive HPyV6 in the healthy skin was 70, clinical grading I. Regarding the three patients negative for MCPyV in non-lesion biopsies the age was 75, 82, and 72 years old, respectively. The patients 75 and 82 years old were classified as clinical grading I, while the 72 years old was clinical grading II.

Comment 12. Discussion. Authors provide a very brief discussion of their research findings. Strengths and limitations are not discussed at all. There are also alternative explanations related to methodology of their findings.

Response: we agree with the reviewer that additional parameters should be tested to conclusively establish a role of MCPyV, HPyV6, HPyV7, and TSPyV in AK. We have added the following text at the end of the Results and Discussion section:

The strength of this work is that the presence of DNA from the four most common dermatropic HPyVs was examined in lesion and non-lesion skin biopsies by the highly sensitive PCR technique and the amplified products were sequenced. A caveat of this study is the limited number of individuals that was investigated. Moreover, there was a bias in the gender because all patients were males. Future research to establish a possible role of MCPyV, HPyV6, HPyV7, and TSPyV in AK should include investigation of the genome copy number, the state of the viral genome, and the expression of viral genes.  In addition, longitudinal studies are needed to establish whether persistent infection is required and whether sequence variations in the viral genomes occur.

Round 2

Reviewer 2 Report

Authors adequately answered comments.